

# Functional diversity of microbial communities in pristine aquifers inferred by PLFA- and sequencing -based approaches

Valérie F. Schwab[1,2*]; Martina Herrmann[4,5], Vanessa-Nina Roth[3]; Gerd Gleixner[3], Robert Lehmann[1], Georg Pohnert[2], Susan Trumbore[3], Kirsten Küsel[4,5], Kai. U. Totsche[1]

[1] Friedrich Schiller University, Institute of Geosciences, Jena, Germany

[2] Friedrich Schiller University, Institute for Inorganic and Analytical Chemistry, Jena, Germany

[3] Max-Planck-Institute for Biogeochemistry, Jena, Germany

[4] Friedrich Schiller University, Institute of Ecology, Jena, Germany

[5] German Centre for Integrative Biodiversity Research (iDiv), Halle-Jena-Leipzig, Leipzig, Germany

* Corresponding author. *E-mail address:* vf.schwab@uni-jena.de (Valérie F. Schwab)

Abstract: Microorganisms in groundwater play an important role in aquifer biogeochemical cycles and water quality. However, the mechanisms linking the functional diversity of microbial populations and the groundwater physicochemistry are still not well understood due to the complexity of interactions between surface and subsurface. Here, we used phospholipid fatty acids (PLFAs) relative abundances to link specific biochemical markers within the microbial communities to the spatio-temporal changes of the groundwater physicochemistry. PLFAs were isolated from groundwater of two physicochemically distinct aquifer assemblages in central



Germany (Thuringia). The functional diversities of the microbial communities were mainly
correlated with groundwater chemistry, including dissolved $O_2$, $Fe_t$ and $NH_4^+$ concentrations.
Abundances of PLFAs derived from eukaryotes and potential nitrite oxidizing bacteria
(11MeC16:0 as biomarker for *Nitrospira moscoviensis*) were high at sites with elevated $O_2$
concentration where groundwater recharge supplies both bioavailable organic substrates and
$NH_4^+$ needed to sustain heterotrophic growth and nitrification processes. In anoxic groundwaters
more rich in $Fe_t$, PLFAs abundant in sulphate reducing bacteria (SRB), iron-reducing bacteria
and fungi increased with $Fe_t$ and $HCO_3^-$ concentrations suggesting the occurrence of active iron-
reduction and the possible role of fungi in meditating iron solubilisation and transport in those
aquifer domains. In $NH_4^+$ richer anoxic groundwaters, anammox bacteria and SRB- derived
PLFAs increased with $NH_4^+$ concentration further evidencing the dependence of the anammox
process on ammonium concentration and potential links between SRB and anammox bacteria.
Additional support of the PLFA-based bacterial communities was found in DNA and RNA-based
Illumina MiSeq amplicon sequencing of bacterial 16S rRNA genes, which evidenced high
predominance of nitrite-oxidizing bacteria *Nitrospira* e.g. *Nitrospira moscoviensis* in oxic zones
of the aquifers and of anammox bacteria in $NH_4^+$ richer anoxic groundwater. Higher relative
abundances of sequence reads in the RNA-based data sets affiliated with iron-reducing bacteria in
$Fe_t$ richer groundwater supported the occurrence of active dissimilatory iron-reduction. The
functional diversity of the microbial communities in these biogeochemically distinct groundwater
assemblages can be largely attributed to the redox conditions linked to changes in bioavailable
substrates and input of substrates with the seepage. Our results demonstrate the power of
complementary information derived from PLFA-based and sequencing-based approaches.
**1.  Introduction**



Continental and marine subsurface environments represent the largest habitat on Earth for
microbial life and therefore are of primary importance for energy fluxes on a global scale
(Edwards et al., 2012). In terrestrial ecosystems, complex interactions between the surface and
subsurface compartments, including aquifers, such as groundwater recharge and rainfall event-
driven flow, influence the availability of $O_2$, and the nature and abundance of bioavailable
organic matter (OM) (Benner et al., 1995; Kalbus et al., 2006). Recent groundwaters tend to
maintain the chemical characteristics of surface, i.e. higher $O_2$ levels and greater amounts of
bioavailable substrates (e.g. labile OM) which support aerobic heterotrophic microbial activity
(Landmeyer et al., 1996). In contrast, ancient groundwaters tend to reflect the chemistry of the
aquifer materials. They have typically lower concentrations of $O_2$ and bioavailable substrates
which cause facultative anaerobes to switch to terminal electron acceptors with lower energy
yield such as $NO_3^-$, $MnO_2$, $FeOOH$ and $SO_4^2$ (Chapelle and Lovley, 1992). In pristine aquifers
low amount of OM typically results in a higher amount of terminal electron acceptors than
electron donors (Chapelle, 2001). As many chemolithoautotrophs can use a variety of compounds
to meet their energy needs in the dark subsurface, increasing numbers of studies report an
important chemolithoautotrophy in groundwater (Stevens and McKinley, 1995; Emerson et al.,
2015; Herrmann et al., 2015). However, how exactly the composition and function of microbial
communities in groundwater depend on hydrology, chemistry and the relationship to groundwater
recharge dynamics is still not well understood.
There are a number of ways to assess the composition and function of microbial
communities in groundwaters. Phospholipid fatty acids (PLFAs) are membrane constituents of all
living organisms. Because various PLFA structures are indicative of specific types or groups of
bacteria in soil (Frostegård and Bååth, 1996; Frostegård et al., 2011) and aquifers (Green and
Scow, 2000), PLFA-based studies are recognised as a valuable approach to infer the presence of



specific microbial groups and to show trends in the spatial distribution of active microbial
populations related to specific substrate utilization patterns in environments (Torsvik and Øvreås,
2002; Schneider et al., 2012). PLFAs commonly associated to a group or genus of bacteria are
branched PLFAs (iso, anteiso) for gram-positive bacteria, mono-unsaturated PLFAs for gram-
negative bacteria, br17:1 (especially i17:1ω7) for *Desulfovibrio* (Edlund et al., 1985; Kohring et
al., 1994), 10Me16:0 for *Desulfobacter* (Dowling et al., 1986; Macalady et al., 2000) and 17:1
(especially 17:1ω6) for *Desulfobulbus* (Parkes and Graham Calder, 1985; Macalady et al., 2000).
Additionally, the PLFAs 18:2ω6,9 and 18:1ω9 are abundant in fungi (Frostegård and Bååth,
1996) and the 20:4, 20:5 and 22:5 and 22:6 PLFAs are common in protozoa (White, 1988) or
algae (Volkman et al., 1989). However, a definitive identification of the lipid sources is often
complicated, because some of those fatty acids may also be found, albeit in smaller amounts, in
cell membranes of other organisms (Frostegård et al., 2011). A few PLFAs are highly specific,
for example ladderanes are characteristic membrane constituents of anammox bacteria
(Sinninghe Damsté et al., 2005; Sinninghe Damste et al., 2002) and have commonly been used to
infer the presence of active anammox bacteria in diverse environments (Kuypers et al., 2003;
Jaeschke et al., 2009). As these organisms are capable of anaerobically oxidizing ammonium
with nitrite to molecular $N_2$, they play an essential role in N removal from marine (Dalsgaard et
al., 2003; Burgin and Hamilton, 2007) and lacustrine environments (Yoshinaga et al., 2011). Yet,
their role in aquifer environments is only starting to be considered (Humbert et al., 2009).

The stable carbon isotope ratios ($\delta^{13}$C values) of PLFAs reflect a combination of the source

of microbial carbon and kinetic isotope fractionation effects associated with the carbon
assimilation pathway (e.g., heterotrophy, autotrophy, methanotrophy; (Teece et al., 1999; Zhang
et al., 2003; Londry et al., 2004). Although a wide range of carbon isotope effects have been
measured, in general autotrophs are expected to have PLFA $\delta^{13}$C values more negative than



heterotrophs (Blair et al., 1985; Teece et al., 1999; van der Meer et al., 2001; Zhang et al., 2003;
Londry et al., 2004; Schouten et al., 2004). In particular, large isotope effects have been
associated with anammox bacteria that have PLFA $\delta^{13}$C values as much as 47 ‰ more negative
than the dissolved inorganic carbon (DIC) source (Schouten et al., 2004).
In this study, we took advantage of the Hainich Critical Zone Exploratory (Hainich CZE;
Küsel et al., 2016), which provides the infrastructure for sampling groundwaters with very
different redox conditions and water chemistry (Kohlhepp et al., 2016). We used PLFA
distributions and $\delta^{13}$C signatures in groundwater within two superimposed pristine carbonate-
rock aquifer assemblages to explore how active microbial communities reflect hydrochemical
changes of the groundwater and its relationships with surface recharge environments.
Additionally, Illumina MiSeq amplicon sequencing targeting 16S rRNA genes and transcripts,
providing a more detailed insight into bacterial community structure and taxonomic affiliation
(Kozich et al., 2013), was used to confirm microbial community structure and potential function
assessed by PLFAs. This study provides baselines for future studies investigating the impact of
changes in surface conditions on microorganism in carbonate-rock aquifer ecosystems.
Additionally, in Schwab et al. (submitted) the $^{14}$C- and $^{13}$C- contents of PLFAs and potential
microbial C sources provide further insights into the heterogeneity of microbial C cycling and
thus contribute to a better understanding of chemoautotrophic versus heterotopic metabolisms
within these aquifer systems.
**2.    Study sites**





*2.1.  Geology*

The sampled groundwater wells are part of the monitoring well transect of the Hainich CZE

(North-western Thuringia, central Germany) of the Collaborative Research Centre (CRC)
AquaDiva. This CRC is devoted to determine how deep signals of surface environmental
conditions can be traced in the critical zone (Küsel et al., 2016). The wells access two distinct
aquifer assemblages in the Upper Muschelkalk (mo) lithostratigraphic subgroup (German
Triassic, Middle Triassic epoch) at different depths and locations (Figure 1). The lower aquifer
assemblage (subsequently referred to as HTL), encountered at depths ranging from 41 m to 88 m
below the surface, is rich in $O_2$, whereas the upper aquifer assemblage (referred to as HTU),
found at depths from 12 m to 50 m below surface, is anoxic to sub-oxic. Both aquifer
assemblages are found in alternating sequences of limestones and marlstones that are partly
karstified (Kohlhepp et al., 2016). More details on the CZE and well constructions can be found
in Küsel et al. (2016) and Kohlhepp et al. (2016). The HTU, comprising several aquifers and
aquitards, is hosted in marine sediments of the Meißner formation (moM) and Warburg formation
(moW) of the Upper Muschelkalk at locations H3 to H5 (Figure 1). The HTL comprises one
aquifer hosted in the Trochitenkalk formation (moTK).
**3.  Methods**
*3.1.  Groundwater sampling*

Groundwater was sampled for chemical analyses and colloidal/particulate organic matter

(POM) in June, September and December of 2014 (Table 1) during regular sampling campaigns
within the coordinated joint monitoring program of the CZE. Groundwater samples were





collected at locations H3, H4, and H5 (i.e. the lower topographic positions of the well transect,
Figure 1). Wells H3.2, H4.2, H4.3, H5.2 and H5.3 reach into the HTU, while wells H3.1, H4.1
and H5.1 access the HTL aquifers (Figure 1). The wells were originally drilled between 2009 and
2011, and were specifically designed sampling groundwater (micro)-organisms and particles.
Prior to sampling, stagnant water (at least three well volumes) was pumped out and discarded
until the physicochemical parameters pH, dissolved $O_2$ concentration, redox potential and
specific electrical conductivity remained constant. Subsequently, ~1000 L of groundwater were
filtered on site using a submersible pump (Grundfos SQ5-70, Grundfos, Denmark) connected to a
stainless steel filter device (diameter 293mm Millipore USA) equipped with a removable pre-
combusted (5 h at 500 °C) glass fiber filter (Sterlitech, USA) of fine porosity (0.3 µm) allowing a
water flow of ca. 20 Lmin$^{-1}$. Filters with the collected particulates were carefully removed and
immediately stored at -80°C until analysis. Groundwater extraction temperature, redox potential,
specific electrical conductivity, pH and dissolved $O_2$ concentration were monitored continuously
during pumping in a flow-through cell equipped with the probes TetraCon 925, FDO 925, Sentix
980, ORP 900 (WTW GmbH, Germany) and meter (Multi 3430 IDS, WTW GmbH, Germany).
During the sampling campaign of June 2014, groundwater was additionally sampled for
nucleic acid extraction. The groundwater was transferred to sterile glass bottles and kept at 4°C.
Within a few hours after sampling, five to six liters of groundwater were filtered through 0.2 µm
pore size polyethersulfone Supor filters (Pall Corporation, USA), and 2 litres were filtered
through 0.2 µm pore size polycarbonate filters (Nuclepore, Whatman, United Kingdom) for
extraction of DNA and RNA, respectively. Filters were immediately transferred to dry ice and
stored at -80°C until nucleic acid extraction.





### 3.2. Groundwater chemistry analyses


Concentration of the major anions ($SO_4^{2-}$, $Cl^-$, $NO_3^-$, $PO_4^{3-}$; PES filter <0.45 μm) were
determined according to DIN EN ISO 10304-1 (2009a) using an ion chromatograph (DX-120,
DIONEX, USA; equipped with an IonPac AS11-HC column and an IonPac AG11-HC pre-
column). The redox sensitive parameters ($Fe^{2+}$, $NO_2^-$, $NH_4^+$) were determined by colorimetry
according to manufacturer's protocol following APHA (1981) and Reardon et al. (1966). The
concentration of DOC and DIC (filter <0.45 μm) were determined by high temperature catalytic
oxidation (multi 18 N/C 2100S, Analytik Jena, Germany) according to DIN EN 1484 (1997).
Total S ($S_t$), Mn ($Mn_t$) and iron ($Fe_t$) were analysed by ICP-OES (725 ES, Varian/Agilent, USA)
according to DIN EN ISO 11885 (2009b). The acid and base neutralizing capacity (ANC, BNC)
by acid/base endpoint-titration was determined according to DIN 38409-7 (2005). The
approximated concentrations of $HCO_3^-$ and $CO_2^-$ were converted from $ANC_{4.3}$ and $BNC_{8.2}$ by
simple replacement ($cCO_2$ (mmolL$^{-1}$)=$BNC_{8.2}$(mmolL$^{-1}$); $cHCO_3^-$ (mmolL$^{-1}$)=$BNC_{4.3}$ (mmolL$^{-1}$)),
assuming that other buffering species than those are negligible, in the nearly pH-neutral waters
(Wisotzky, 2011).

### 3.3. PLFA extraction and pre-treatment


PLFAs were extracted from filters using a method slightly modified from the described
by Bligh and Dyer (1959) and Seifert et al. (2013). The filters were cut into small pieces and
extracted in a phase solution of chloroform-methanol (2:1; v/v) with 0.005 M phosphate buffer.
The solution was rotated and shaken for 4 h. Chloroform and water (1:1; v/v) were then added to
the mixture. After shaking, the chloroform phase, containing the total lipid extract (TLE), was
separated from the water-MeOH phase and, concentrated by a rotary evaporator. The TLE was



then partitioned into the conventionally defined neutral lipids (NL), glycolipid (GL) and
phospholipid (PL) fractions by chromatography (SPE 6 ml column) on pre-activated silica gel
(Merck silica mesh 230-400, 2 g pre-activated 1h et 100 °C) using chloroform (12 ml), acetone
(12 ml) and methanol (48 ml), respectively. The phospholipids were converted to fatty acid
methyl esters (FAME) using mild-alkaline hydrolysis and methylation (White et al., 1979). The
different fatty acids were then separated using $NH_2$ column (Chromabond 3ml, 500 mg) with 3
ml of hexane/DCM (3:1; v/v) for eluting the unsubstituted FAMEs; 3 ml of DCM/ ethylacetate
(9:1; v/v) for PLOHs and 6 ml of 2% acetic acid in methanol for unsaponifiable lipids. To
quantify the recovery, the standard, 1,2-dinonadecanoyl-sn-glycero-3-phosphatidyl-choline
(Avanti Polar Lipids, Inc. USA), was added on a clean pre-combusted glass filter that was treated
exactly as the samples following the above protocol. The formed C17:0 FAME was quantified to
calculate a mean recovery of 82%.
*3.4.   Nucleic acid extraction, amplicon sequencing, and sequence analysis*
DNA was extracted from the polyethersulfone filters using the Power Soil DNA
extraction kit (Mo Bio, CA, USA) following the manufacturer's instructions. RNA was extracted
from polycarbonate filters using the Power Water RNA Isolation Kit (Mo Bio, CA, USA). Traces
of co-extracted genomic DNA were removed using Turbo DNA free (Thermo Fisher Scientific,
Germany), and reverse transcription to cDNA was performed using ArrayScript Reverse
Transcriptase (Thermo Fisher Scientific) as described previously (Herrmann et al., 2012). DNA
and cDNA obtained from the groundwater samples from PNK51 were shipped to LGC Genomic
GmbH (Berlin, Germany) for Illumina MiSeq amplicon sequencing of the V3-V5 region of 16S
rRNA genes and transcripts, using the primer combination Bakt_314F/Bakt_805R (Herlemann et
al., 2011). Sequence analysis was performed using Mothur v. 1.36 (Schloss et al., 2009),





following the MiSeq SOP (http://www.mothur.org/wiki/MiSeq_SOP; Kozich et al., 2013).
Quality-trimmed sequence reads were aligned to the SILVA reference database (v 119; Quast et
al., 2013). Potential chimeric sequences were detected and removed using the uchime algorithm
implemented in Mothur. Taxonomic classification of sequence reads was based on the SILVA
reference database (v 119). To facilitate comparisons across samples, sequence read numbers per
sample were normalized to the smallest number of sequence reads obtained across all samples
using the subsample command implemented in Mothur. Raw data from 16S rRNA amplicon
Illumina sequencing were submitted to the European Nucleotide Archive database under the
study accession number PRJEB14968 and sample accession numbers ERS1270616 to
ERS1270631.
*3.5.  Gas chromatography and gas chromatography-mass spectrometry*
Ten percent of the PLFA extracts were used for peak identification and quantification
using a gas chromatograph (Trace 1310 GC) coupled to a triple quadrupole mass spectrometer
(TSQ-8000; Thermo-Fisher, Bremen, Germany) at the Friedrich Schiller University Jena,
Institute of Inorganic and Analytical Chemistry (Germany). The GC was equipped with a TG-
5silms capillary column (60 m, 0.25 mm, 0.25-μm film thickness). Helium was used as carrier
gas at a constant flow of 1.2 ml min$^{-1}$, and the GC oven was programmed to have an initial
temperature of 70 °C (hold 2 min), a heating rate of 11°C min$^{-1}$, and a final temperature of 320
°C, held for 21 min. The PTV injector was operated in splitless mode at an initial temperature of
70 °C. Upon injection, the injector was heated to 300 °C at a programmed rate of 720 °C min$^{-1}$
and held at this temperature for 2.5 minutes. FAMEs were quantified relative to an internal
standard nonadecanoic acid-methyl ester (19:0) added prior to GC analysis and relative to a
standard mixture (FAME-Mix, Thermo-Fisher, Bremen, Germany) measured in 5 different



concentrations between 2 and 40 ng/µl. FAMEs were identified based on the mass spectra and on
retention time of standards. Standard nomenclature is used to describe PLFAs. The number
before the colon refers to the total number of C atoms; the number(s) following the colon refers
to the number of double bonds and their location (after the ′ω') in the fatty acid molecule. The
prefixes "me," "cy," "i," and "a" refer to the methyl group, cyclopropane groups, and iso- and
anteiso-branched fatty acids, respectively.
*3.6. PLFA distribution and statistical analyses*
The concentrations of forty-seven PLFAs, expressed in mol %, were investigated in the different
wells (Supplement Table S1). The sum of the PLFAs considered to be predominantly of bacterial
origin (BactPLFA; i15:0, a15:0, 15:0, 16:1ω7, 16:0, cy17:0, 18:1ω7, 18:0 and cy19:0) was used
as an index of the bacterial biomass (Bossio and Scow, 1998; Frostegård and Bååth, 1996). The
fungal biomass (FunPLFA) was estimated from the sum of the concentrations of the 18:2ω6c
(Bååth et al., 1995), 18:3ω6c (Hamman et al., 2007) and 18:1ω9c (Myers et al., 2001); these were
all significantly correlated with each other. Gram-positive (G+) bacteria were represented by the
sum of PLFAs: i12:0, i13:0, a15:0, i15:0 (Kaur et al., 2005). Gram-negative (G-) bacteria
included 16:1ω7c, cy17:0, 18:1ω7c and cy19:0 (Kaur et al., 2005). The ratios of
FunPLFA/BactPLFA and $G_+$/G- were calculated from the above PLFAs.
The PLFA data in mol % and twenty-nine environmental parameters were used for
principal component analysis (PCA) and redundancy analyses (RDA) using CANOCO for
Windows, version 5 (Microcomputer Power, Ithaca, New York, United States). Before
regression, the data were centered and standardized. We used PCA to emphasise strong variations
and similarities of the PLFA distributions between the wells and identify patterns in the dataset.
RDA is used to determine PLFA variations and similarities (response variables) that can be



significantly explained by different environmental parameters (explanatory variables). This
technique helps to identify the environmental parameters that have the highest effects on the
PLFA distribution, i.e. on the microbial communities in the different wells.

Additionally, we used variation partitioning analyses with conditional effects to determine

the variations in PLFA composition between the different wells that can be explained
significantly by the preselected environmental variables. To visualise the PLFAs acting
significantly with the environmental variables (predictor), we used PLFA-environmental
variables t-value biplots (Šmilauer and Lepš, 2014). These plots can be used to approximate the t-
values of the regression between a particular PLFA and an environmental variable. The PLFAs
are represented by arrows projecting from the origin. Those with a preference for higher values of
the environmental variable are enclosed by a red (indicating positive relationship) circle.
Inversely, those with preference for low values of the corresponding environmental variable have
their arrow-tips enclosed by a blue (indicating negative relationship) circle.
*3.7.  Compound-specific stable isotope carbon measurements*

The carbon stable isotope composition of pre-purified PLFAs were determined using a

GC-C-IRMS system (Deltaplus XL, Finnigan MAT, Bremen, Germany) at the Max-Planck-
Institute (MPI) for Biogeochemistry, Jena. Analyses were performed using 50 % of the total
amount of PLFA extracts. The gas chromatograph (HP5890 GC, Agilent Technologies, Palo Alto
USA) was equipped with a DB1-ms column (60 m, 0.25 mm ID, 0.52 um film thickness,
Agilent). The injector at 280 °C was operated in splitless mode with a constant flow of 1 ml min$^{-1}$.
$^{1}$. The oven temperature was maintained for 1 min at 70 °C, heated with 5 °C min$^{-1}$ to 300 °C and
held for 15 min, then heated with 30 °C min$^{-1}$ to 330 °C and hold 3 min. Isotope values,
expressed in the delta notation (‰), were calculated with ISODAT version software relative to





the reference $CO_2$. Offset correction factor was determined on a daily basis using a reference
mixture of $n$-alkanes ($n$-$C_{17}$ to $n$-$C_{33}$) of known isotopic composition. The carbon isotopic
composition of the reference $n$-alkanes was determined off-line using a thermal conversion
elemental analyser (TC/EA) (Thermo-Fisher, Bremen, Germany) interfaced to the DELTA V
PLUS irMS system via a Conflo III combustion interface (Thermo-Fisher, Bremen, Germany;
Werner and Brand, 2001). The contribution of the methyl carbon derived from the methanol after
mild-alkaline hydrolysis and methylation of the PLFAs to the FAME was removed by isotopic
mass balance, with $\delta^{13}C_{PLFA} = [(N_{PLFA} + 1) \times \delta^{13}C_{FAME} - \delta^{13}C_{MeOH}] / N_{PLFA}$ where N is the number
of carbon atoms in the PLFA and $\delta^{13}C_{FAME}$ stands for the measured values of the methylated
PLFAs (Kramer and Gleixner, 2006). The carbon isotope composition of MeOH used for
derivatisation ($\delta^{13}C$ value = -31.13 ± 0.03) was determined off-line using a thermal conversion
elemental analyzer (TC/EA) (Thermo-Fisher, Bremen, Germany) interfaced to the DELTA V
PLUS irMS system via a Conflo III combustion interface (Thermo-Fisher, Bremen, Germany).
**4.  Results**
*4.1.  Groundwater physicochemistry*

The deeper aquifer assemblage, HTL (wells H3.1, H4.1 and H5.1), had higher mean

concentration of $O_2$ (4.1± 1.2 $mgL^{-1}$) than the shallow aquifer assemblage, HTU (wells H4.2,
H4.3, H5.2 and H5.3). Groundwater extracted from HTU wells were anoxic with $O_2 < 0.02$ $mgL^{-}$
$^1$ (Supplement Table S2 and Figure 2) except for well H3.2 that had mean $O_2 = 2.2 ± 0.5$ $mgL^{-1}$.
No significant differences in the content of dissolved organic carbon (DOC: mean = 0.46 ± 0.2
$mgL^{-1}$) were measured between the different wells. In agreement with more oxic condition, the





HTL had higher mean concentration of nitrate ($10.4 \pm 6.6$ mgL$^{-1}$) and sulphate ($183.4 \pm 110.8$
mgL$^{-1}$) than the anoxic HTU ($5.6 \pm 2.9$ mgL$^{-1}$ and $63.6 \pm 22.2$ mgL$^{-1}$, respectively). Higher mean
concentrations of total iron ($Fe_t = 0.1 \pm 0.08$ mgL$^{-1}$), TIC ($94.7 \pm 7.6$ mgL$^{-1}$) and HCO$_3^-$ ($4.69 \pm$
$0.07$ mgL$^{-1}$), the latter measured as acid neutralizing capacity (Wisotzky, 2011), were found in
the anoxic groundwater of the wells H4.2 and H4.3 than of the wells H5.2 and H5.3 that had
mean $Fe_t = 0.01 \pm 0.00$ mgL$^{-1}$, TIC $= 77.3 \pm 5.4$ mgL$^{-1}$ and HCO$_3^- = 4.02 \pm 0.35$ mgL$^{-1}$ (Figure
2). Inversely, mean concentrations of total sulphur ($S_t = 26.1 \pm 4.9$ mgL$^{-1}$), sulphate ($76.7 \pm 14.8$
mgL$^{-1}$) and ammonium ($0.62 \pm 0.15$ mgL$^{-1}$) were higher in the anoxic groundwater of the wells
H5.2 and H5.3 than of the wells H4.2 and H4.3 that had mean $S_t = 12.3 \pm 6.0$ mgL$^{-1}$, SO$_4^{2-} = 37.4$
$\pm 20.6$ mgL$^{-1}$ and NH$_4^+ = 0.13 \pm 0.06$ mgL$^{-1}$ (Figure 2 and Supplement Table S2).

The PCA analyses using the physicochemical parameters of the groundwater separate the

wells in three main groups (Figure 3) with 73.6% of the variability explained by the first three
principal components (PC): PC1, 32.8%; PC2, 23.8% and PC3, 16.9%. The conductivity, redox
potential and the concentration of Ca$^{2+}$, SO$_4^{2-}$, S$_t$ and O$_2$ positively correlated (response $> 0.5$)
with PC1 separating the oxic to sub-oxic wells H5.1, H4.1, H3.1 and H3.2 from the anoxic wells
H4.2/3 and H5.2/3. The concentrations of NH$_4^+$, K$^+$ and Mg$^{2+}$ inversely correlated (response $<$
0.5) with PC1, separating wells H5.2/3 from the others. The Fe$_t$, TIC and HCO$_3^-$ positively
correlated along PC2 and mainly separated the anoxic wells between location H4 (with higher
iron and DIC concentration but lower NH$_4^+$, SO$_4^{2-}$ and St concentration) and location H5 (with
lower iron and DIC but higher NH$_4^+$, S$_t$, and SO$_4^{2-}$ concentration).
*4.2.   PLFA distribution and statistical analyses*

The 16:1ω7c (mean $23.4 \pm 8.7$ mol %), 16:0 (mean $13.6 \pm 3.3$ mol %) and 18:1ω7c (mean

$6.2 \pm 5.4$ mol %), common in most bacteria, were abundant in both aquifer assemblages



(Supplement Table S1). The PLFAs 10Me16:0 (mean $6.5 \pm 4.5$ mol %), 17:1ω6c (mean $6.5 \pm 4.5$
mol %), 17:1 (mean $0.8 \pm 0.75$ mol %) and iC17:1 (mean $1.2 \pm 1.0$ mol %) derived from
Deltaproteobacteria mainly encompassing SRB, iron-reducing or oxidizing bacteria were
dominant only in the anoxic groundwater, whereas the 11Me16:0 (mean $3.3 \pm 3.6$ mol %) were
found only in the oxic groundwaters. The [3]- and [5]- ladderane PLFAs specific to anammox
bacteria were found in the anoxic wells H5.2 and H5.3 and the sub-oxic well H3.2 in a
concentration of up to 4.3 mol %. The highest fungal biomass, based on the FunPLFA/BactPLFA
ratios (Table 2), was observed in the anoxic wells H4.2 and H4.3 (mean $0.3 \pm 0.2$), whereas the
lowest in the anoxic wells H5.2 and H5.3 (mean $0.03 \pm 0.04$). Additionally, the PLFA 20:4, 20:5,
22:5 and 22:6 were observed in different concentration in all wells. The Gram negative (G-)
bacteria were more abundant than Gram positive bacteria (G+) in both HTU and HTL (Table 2:
mean G+/G- ratio = $0.4 \pm 0.2$). The highest values of the G+/G- ratios were in the anoxic wells
H4.2 and H4.3 (mean $0.7 \pm 0.1$).

A PCA analysis explained 54.4 % of the PLFA variation with 3 principal components;

PC1 explaining 27.8%; PC2, 15.3% and PC3, 11.3% of overall variability (Figure 4). The PCA
analyses of the PLFAs also separated the wells into three main groups. The wells of the upper
aquifer assemblage were separated along PC1; wells from sites H4 separated from those of the
sites H5/H3. Along PC2, the wells were separated between the oxic (well H3.1, H4.1 and H5.1),
sub-oxic (well H3.2) and anoxic groundwater (H4.2, H4.3, H5.2, H.5.3). The RDA analyses
showed that $O_2$, $Fe_t$ and $NH_4^+$ concentrations or $O_2$, $HCO_3^-$ and $NH_4^+$ concentrations explained
the greatest proportion (38%) of the PLFA variability (Figure 5). Well grouping obtained using
the RDA analysis was consistent with the results of the PCA. The first RDA axis (20.2 %)
separated the anoxic wells of the upper aquifer according to $Fe_t$ or $HCO_3^-$ (wells H4.2 and H4.3)
and $NH_4^+$ (wells H5.2 and H5.3) concentration. The second RDA axis (14.0 %) separated suboxic



to oxic (mainly lower aquifer) from anoxic groundwater (upper aquifer). In the following
discussion, the wells are separated according the PCA and RDA analyses into these three groups.

To identify the individual effects of $O_2$, $Fe_t$ and $NH_4^+$ on the explained PLFA variation,

we used variation partitioning with conditional effects implemented in Canoco 5 (Heikkinen et
al., 2004; Roth et al., 2015). Because these environmental variables were the most significant
factors, their combined variation was set to explain 100% of total PLFA variation in each RDA
plot. In our case, the following eight fractions explained the PLFA distribution by effect of $O_2$
alone; a = 19.7%, effect of $NH_4^+$ alone; b = 22.0%, effect of $Fe_t$ alone; c = 13.4%, and by
combined effects of $O_2$ and $NH_4^+$; d = 22.3%, by combined effects of $Fe_t$ and $NH_4^+$; e = 29.2%,
and by combined effect of $O_2$ and $Fe_t$; f = 25.9%. The fraction g (-32.4%) explained the
combined effect of the three environmental variables (Figure 6). The PLFA-environmental
variable $O_2$ t-plot (Figure 6A) showed that mol % concentration of Me15:0, 16:1ω11c, cy17:0,
11Me16:0, 18:1 and 22:6 increased significantly with $O_2$ concentration whereas 10Me12:0, i13:0,
a15:0, 17:1, i17:1 and [5]-ladderane mol % concentration decreased with $O_2$ concentration. The
PLFA-environmental variable $Fe_t$ t-values biplot (Figure 6B) showed that 10Me12:0, 16:1,
18:1ω9c, 18:1ω7c, i17:1 and cy19:0 mol % concentration increased with $Fe_t$ concentration,
whereas 10MeC16:0, 17:1, [3]-ladderane and [5]-ladderane mol % decreased. Inversely, the
PLFA-environmental variable $NH_4^+$ t-values biplot (Figure 6C) showed that 10Me16:0, 17:1, [3]-
ladderane and [5]-ladderane mol % concentration increased with $NH_4^+$ concentration, whereas
10Me12:0, 16:1, 18:1ω9c, 18:1ω7c, i17:1 and cy19:0 mol % concentration decreased.
*4.3.  PLFA $\delta^{13}C$ values*

The PLFA $\delta^{13}C$ values for individual compounds ranged from -26‰ to - 68.8‰

(Supplement Table S3 and Figure 7). The most negative mean $\delta^{13}C$ values were found in the





anoxic groundwater from location H5.2 and H 5.3 (-48.0 ±10.5‰ and -45.9 ±11.7‰,
respectively) and in the suboxic groundwater at the location H3.2 (-45.4‰ ±9.0) and coincided
with the presence of the [5]- and [3]-ladderane. In those wells, the i13:0 (-52.4 ± 2.0‰), i15:0 (-
55.6 ± 2.0‰), 10Me16:0 (-56.1 ±2.1‰) and i17:1 (-44.3 ± 2.0‰) were slightly $^{13}$C-depleted
compared to both [5]- and [3]-ladderane (-65.6 ± 2.0‰). More positive mean PLFA $\delta^{13}$C values
were measured in the anoxic wells H4.2 and H4.3 (-36.8‰ ± 2.1) and in the oxic wells H5.1,
H4.1 and H3.1 (-35.3‰ ± 1.1). In those wells, the $\delta^{13}$C values of the i13:0, i15:0 and 10MeC16:0
were in the same range as the other PLFA (Figure 7). The most positive $\delta^{13}$C values were
measured for 16:1ω11c and 11MeC16:0 in the oxic wells H5.1 and H4.1 (mean -28.2‰ ± 2.5)
and for 18:1ω9c (mean -30.2‰ ± 2.3) in the anoxic wells H4.2 and H4.3.
*4.4.  Bacterial community composition based on 16S rRNA gene sequences*
Based on Illumina sequencing of DNA-based 16S rRNA gene amplicons, bacterial
communities were largely dominated by members of the phylum Nitrospirae and of Candidate
Division OD1, followed by Delta- and Betaproteobacteria, Planctomycetes, Alpha- and
Gammaproteobacteria (Figure 8A). Members of the Nitrospirae were especially abundant in the
groundwater of the anoxic wells H5.2 and H5.3 as well as the oxic wells H4.1 and H5.1, while
this phylum only contributed a minor fraction in the groundwater of the anoxic wells H4.2 and
H4.3 and the oxic wells H3.1 and H3.2 (Figure 8A). In addition, we performed sequencing of 16S
rRNA amplicons derived from the extracted RNA to get insight into which taxonomic groups
harbor protein synthesis potential as proposed by Blazewicz et al. (2013). RNA-based community
analysis targeting 16S rRNA sequences has traditionally been used as an approximation of the
currently active fraction of the microbial community. However, this interpretation is critical since
many cells may retain high ribosome contents even in a dormant state (Filion et al., 2009;



Sukenik et al., 2012) and thus, rRNA content of cells does not necessarily indicate current
metabolic activity, especially in low-nutrient environments such as groundwater (reviewed in
Blazewicz et al., 2013). Here, we used this approach to investigate whether key microbial groups
identified by PLFA-based analysis were supported to be metabolically active or have the
potential to resume metabolic activities based on the detection of the corresponding 16S rRNA
gene sequences on the RNA level. In general, members of the Candidate Division OD1 formed
only a minor part of the community obtained by RNA-based amplicon sequencing while
members of the phyla Nitrospirae, Planctomycetes, and Proteobacteria showed the largest relative
abundances (Figure 8B). Members of the phylum Nitrospirae were especially highly represented
in the RNA-based analyses of wells H3.2, H4.1, and H5.2 and H5.3. Among the Proteobacteria,
Deltaproteobacteria were more frequently represented in the RNA-based analysis of communities
of wells H3.1, H3.2, H5.2, and H5.3 while Alphaproteobacteria showed a higher relative
abundance in the groundwater of wells H4.2, H4.3 and H5.1 (Figure 8B).
Bacterial phyla and classes may harbor organisms with a high diversity of different
metabolisms. Therefore, as some source specific PLFA displayed strong relationships with the
environmental variables $O_2$, $NH_4$, and $Fe_t$, we specifically focused on groups potentially involved
in iron oxdiation and reduction, sulfate reduction, anammox, and nitrite oxidation. Here, relative
fractions of reads assigned to bacterial genera known to be involved in either of these processes
were summed up to get an estimation of the potential for these processes within the microbial
community with both DNA- and RNA-based analyses. On the level of DNA-based sequencing,
bacteria involved in iron oxidation accounted for 0.25 to 6.2% of the sequence reads across sites
(Figure 9A) while they accounted for 0.24 to 2.8% on the level of the RNA-based analyses with
the highest relative fraction of bacteria potentially involved in iron oxidation at wells H5.1 and
H5.3 (Figure 9B). Differences across sites and aquifers were more pronounced for bacteria





involved in iron reduction, which were accounted for by 0.16 to 3.7% of the sequence reads on
the DNA level but for 0.15 to 20.4% on the RNA level with the highest number of sequence
reads affiliated with known iron reducers in the groundwater of well H4.3 (Figure 9B) Bacteria
related to the genera *Acidiferrobacter*, *Gallionella*, and *Sideroxydans* were the most frequent
genera among the known iron oxidizers while members of the genera *Albidiferax* and
*Ferribacterium* dominated the iron reducing groups. Bacterial groups potentially involved in
sulfur reduction included the genera *Desulfacinum*, *Desulfovibrio*, *Desulfosporosinus*,
*Desulfatiferula* as the most frequent groups and accounted for 0.2 to 2.8% of the sequence reads
on the DNA level and 0.4 to 10.4% on the RNA level with the maximum in the anoxic well H4.2
(Figure 9). Anammox bacteria mostly represented by the Candidatus genera *Brocadia* and
*Kuenenia* accounted for 0.6 to 3.0% of the sequence reads on the DNA level and for 1.1% to
16.8% on the RNA level with the highest fractions in the groundwater of the wells H3.1, H5.1,
H5.2 and H5.3 (Figure 9). Finally, we observed large fractions of potential nitrite oxidizers
mostly related to the genus *Nitrospira* with the vast majority of the *Nitrospira*-affiliated reads
especially in the lower aquifer assemblage showing a high sequence similarity to the 16S rRNA
gene sequence of *Nitrospira moscoviensis* (96 - 99%). Moreover, reads associated with the genus
*Nitrospira* may also include potential comammox organisms (Pinto et al., 2016). Relative
fractions of sequence reads affiliated with this genus on the DNA and RNA level were highest in
the oxic groundwater as the well H4.1 and lowest in the anoxic groundwater of wells H4.2 and
H5.2 (Figure 9). Since nitrifiers such as *Nitrospira* are known to retain a high ribosome content
even if cells are not active (Morgenroth et al., 2000), these results do not necessarily indicate high
nitrite oxidation activity at the time point of sampling but point to nitrite oxidizers forming a
large fraction of the microbial community with protein synthesis potential.



## 5. Discussion

### 5.1. PLFAs distribution

The PCA of PLFAs indicated that the oxic/suboxic and anoxic groundwaters had distinct bacterial communities, with the anoxic groundwater additionally differentiated into two distinct bacterial communities (Figure 4). Of the environmental variables tested, the variation partitioning showed that $NH_4^+$, $O_2$ and Fet concentration explained 22.0%, 19.7% and 13.4% of the PLFA variations, respectively (Figure 6), and differentiated those three bacterial communities. Variation partitioning analyses revealed, along those environmental variables, clusters of covarying PLFAs that may originate from the same functional group of organisms or closely affiliated organisms that react similarly to certain environmental conditions. While the ladderanes are unequivocally attributed to anammox bacteria (Sinninghe Damsté et al., 2005; Sinninghe Damste et al., 2002), the other PLFAs are not exclusive to a phylogenetic or functional microbial group which complicates their use to understand the role of microbes in environments. The t-value biplots of variation partitioning analyses evidenced the PLFAs that significantly correlated with the environmental variables $O_2$ (Figure 6A) $Fe_t$ (Figure 6B) and $NH_4^+$ (Figure 6C), and provided better insights into the functional diversity of active microorganisms in the subdivided groundwaters. Additional supports of the bacterial community structure, assessed by the PLFA patterns, were found in the 16S rRNA-based results. Although a large fraction of the microbial community remains poorly classified and thus precludes the knowledge of the metabolic capacities, high sequence similarity to genera known to be involved in iron oxidation or reduction, sulphate reduction, anammox and nitrite oxidation allowed an estimation of the fraction of the microbial population potentially involved in these processes. By combining the PLFA-based and sequencing-based approaches, we aimed, here, to compensate for biases





introduced by PCR as well as for the limited phylogenetic resolution of PLFA-based analysis.
This combined approach resulted in highly supported evidences of some key microbial players
and associated biogeochemical processes in physicochemical distinct aquifer assemblages of the
aquifer transect.
*5.1.1. PLFA cluster in oxic to suboxic groundwater (wells 5.1, 4.1, 3.1 and (3.2))*

A cluster of the covarying 20:4, 20:5, 22:5 and 22:6 PLFAs has to our knowledge heretofore

never been observed in groundwater. Associations of those PLFAs have been commonly found in
eukaryotes as microalgae (Volkman et al., 1989), fungi (Kennedy et al., 1993; Olsson, 1999),
particularly ectomycorrhizal fungi (Shinmen et al., 1989), higher plants (Qi et al., 2004) and
protozoans (White, 1988). Protozoa act as detritivores and are expected to be key predators in the
microbial loop feeding on different subsets of the bacterial communities and other protozoa (Brad
et al., 2008; Akob and Küsel, 2011). Consistently, sessile and free swimming suspension feeding
flagellates, e.g., *Spumella* sp., mobile naked amoebae and ciliates could be detected in this
aquifer with a cultivable protist abundance of up to 8.000 cells $L^{-1}$ (Risse-Buhl et al., 2013). 18S
rRNA gene sequences also revealed high relative fractions of *Spumella*-like Stramenopiles, and
sequences affiliated with fungi and metazoan grazers. DNA based pyro-tag sequencing of fungal
internal transcribed spacer (ITS) sequences revealed a fungi community structure dominated by
Ascomycota and Basidiomycota (Nawaz et al., 2016) with the majority of the observed fungal
groups being involved in ectomycorrhizal symbioses. In general, the abundance of micro-
eukaryotes in pristine groundwater is estimated to be low, because they are limited in nutrients,
space, and are unable to cope with oxygen limitations (Akob and Küsel, 2011). Consistently, they
are commonly found in higher concentrations in OM-rich contaminated groundwaters (Ludvigsen
et al., 1997). In pristine aquifers, the origin of those eukaryotic organisms is difficult to determine
as they may be autochthonous, allochthonous or both. In the studied sites, the close relation of



481 eukaryotic PLFA biomarkers with $O_2$ concentrations (Figure 6A) suggests their association with

482 recharging groundwater within larger conduits prone to faster water flow. Freshly introduced

483 surface OC and $O_2$ could fuel the heterotrophic bacterial growth in groundwater. This may

484 subsequently stimulate protists that selectively graze on the prokaryotic biomass and result in the

485 observed relationship between the eukaryotic PLFAs and the $O_2$ concentration. It is possible to

486 speculate that some surface microorganisms would also survive the transport from surface to the

487 aquifer (Dibbern et al., 2014), especially if the transport is fast. In this case, high cy17:0 to

488 16:1ω7c ratios (Table 2) may evidence physiological stress due to change of the environmental

489 conditions within the gram negative communities (Balkwill et al., 1998).

490  The 16:1ω11c and particularly the 11MeC16:0 are major components of *Nitrospira*

491 *moscoviensis* (Lipski et al., 2001) cell membranes, an obligatory chemolithoautotrophic nitrite-

492 oxidizing bacterium (NOB: Ehrich et al., 1995). In the oxic groundwater, the occurrence of 16S

493 rRNA gene sequence reads closely related to *Nitrospira moscoviensis* (Herrmann et al., 2015)

494 further supports the potential of 11MeC16:0 as biomarker for *Nitrospira moscoviensis* and

495 confirms previous assumptions about an important role of nitrite oxidizers within the autotrophic

496 community of the lower aquifers (Herrmann et al., 2015). The correlation of 11MeC16:0 and

497 16:1ω11c with $O_2$ (Figure 6A) indicated the occurrence of active nitrification in oxic zones of the

498 aquifers in agreement with observation of experiments (Satoh et al., 2003). *Nitrospira* use the

499 reverse tricarboxylic acid cycle as the pathway for $CO_2$ fixation (Lücker et al., 2010) which leads

500 to small $^{13}C$ fractionation (2 - 6‰) between biomass and $CO_2$ (van der Meer et al., 1998). The

501 $^{13}C$-enrichment of 11MeC16:0 and 16:1ω11c relative to the other PLFAs (up to 18‰ in well

502 H4.1) supports thus major *Nitrospira* contribution to those PLFAs found in oxic groundwaters

503 (Figure 7).



*5.1.2.   PLFA cluster in anoxic $Fe_t$ richer groundwater (wells H4.2 and H4.3)*
In groundwater the concentration of dissolved iron is often inversely related to oxygen as $O_2$
in water will chemically oxidize iron that will precipitate as insoluble iron-hydroxides at neutral
pH. In the wells H4.2/4.3, the increase of the PLFAs 10MeC12:0, 16:1, 17:1, 18:1ω7c, 18:1ω9c
and cy19:0 with concentrations of $Fe_t$, $Fe_2^+$ and $HCO_3^-$ (Figure 5 and 6B) and the DNA- and
RNA-based analyses  (Figure 9)  suggested degradation of OM by anaerobic iron-reducing
bacteria. Because many iron-reducing bacteria are highly versatile, i.e. they can use different
metal substrates as electron acceptors coupled to the oxidation of the OM (Coleman et al., 1993;
Lovley et al., 1993; Holmes et al., 2004), specific PLFAs linked to the reduction of iron in
anoxic environments are poorly described. The two most studied genera of IRB are *Geobacter*
and *Shewanella* which contain most of those PLFAs (Coleman et al., 1993; Lovley et al., 1993;
Hedrick et al., 2009).  However none of these PLFAs are specific to a certain genus or species.
The 17:1 and cy19:0 are generally related to anaerobic SRB (Dowling et al., 1986) as
*Desulfobulbus* (Parkes and Graham Calder, 1985; Macalady et al., 2000) but also occur in
dissimilatory iron-reducing bacteria as *Shewanella* (Coleman et al., 1993). The ability of some
sulphate reducers to reduce iron rather than sulphate has long been recognized in groundwater
(Coleman et al., 1993).
The 18:1ω9c is common and abundant in fungi (Frostegård and Bååth, 1996; Kaiser et al.,
2010), but may also occur in micro-algae (Arts et al., 2001) and gram-negative bacteria
(Kandeler, 2007). The 18:1ω9c, 18:2ω6,9 and 18:3ω6 are typically used as fungi biomarkers in
soil (Frostegård and Bååth, 1996; Bååth and Anderson, 2003; Ruzicka et al., 2000) and more
particularly for saprotrophs (Etingoff, 2014). The correlations between 18:1ω9c, 18:2ω6,9 and
C18:3ω6 suggested a major fungal origin of those PLFAs in the studied groundwaters. In soil,
fungi are well known for their role in accelerating weathering and solubilisation of iron-





containing minerals by excreting organic acids including phenolic compounds, siderophores,
and protons (Arrieta and Grez, 1971; Landeweert et al., 2001). By forming dense hyphae
tunnelling in soils and shallow rocks, fungi mediate and facilitate iron transport in plants and
increase iron availability in the environment (van Schöll et al., 2008). Therefore, several studies
have linked enhanced rates of iron cycling to the presence of fungal biomass (Gadd, 2010).
Moreover, in a recent study, it is been shown that rhizoplanes are important root channels for
preferential vertical transport from soil to seepage area of soil colloids including microbes
(Dibbern et al., 2014). Limitation of ferric iron may restrain the growth and activity of IRB in
subsurface (O'Neil et al., 2008). In the groundwater of wells H4.2 and H4.3, the close relation of
$18:1\omega9c$ and $18:2\omega6,9$ with $Fe_t$ concentration (Figure 6B) suggested that fungal biomass may,
by mediating and facilitating the transport of different types of organic/inorganic particles and
colloids, play a key role in iron bioavailability and thus sustain IRB growth and activity.
*5.1.3.   PLFA cluster in anoxic $NH_4^+$ richer groundwater (wells H5.2 and H5.3 and (3.2))*

To our knowledge, this is the first time phospholipid [3]-ladderane and [5]-ladderane,

which attest the presence of viable or recently degraded anammox bacteria (Jaeschke et al.,
2009), have been identified in groundwater. The occurrence of anammox bacteria in those
groundwaters is consistent with the DNA- and RNA-based analyses (Figure 9) and coincided
with higher concentrations of ammonium (Figure 2). The difference between DIC and ladderanes
$\delta^{13}C$ values of 55‰ was within the range previously reported for anammox in Black Sea
(Schouten et al., 2004), further suggesting that autotrophic carbon fixation pathways within the
diverse group of anaerobic ammonium-oxidizing bacteria are similar (Schouten et al., 2004). In
the sub-oxic (well H3.2) and anoxic groundwaters (well H5.2 and H5.3), the increasing
concentration of ladderane lipids derived from anammox bacteria with decreasing $O_2$
concentration (Figure 6A) agrees well with the reported high sensitivity of the anammox process



to $O_2$ (Kalvelage et al., 2011). Denitrification and anammox are the dominant nitrogen loss
pathways in aquatic ecosystems (Burgin and Hamilton, 2008; Koeve and Kähler, 2010). The
occurrence of lipids derived from anammox bacteria in those groundwaters indicates that the
anammox process may be critically important in the nitrogen loss from this part of the aquifer
assemblage.
High amounts of 10MeC16:0 are typically found in SRB (Dowling et al., 1986; Vainshtein et
al., 1992; Kohring et al., 1994) but also occur in anammox bacteria (Sinninghe Damste et al.,
2002). Anammox bacteria strongly fractionate against $^{13}$C, producing ladderane lipids which are
$^{13}$C-depleted by 47‰ compared to the inorganic carbon source (Schouten et al., 2004). Relative
to ladderanes, SRB-derived lipids are expected to be $^{13}$C-enriched as cultured SRB under
heterotrophic and autotrophic growth fractionated against $^{13}$C by up 27‰ (Londry et al., 2004).
Therefore, the $^{13}$C-enrichment of 10MeC16:0 (up to 19‰) relative to the ladderanes supported
major SRB contribution to the 10Me16:0 found in these groundwaters. The i13:0, i15:0 and i17:1
are typically, as 10MeC16:0, associated with SRB (Edlund et al., 1985; Kohring et al., 1994). In
those groundwaters, similar $\delta^{13}$C values, in the -44 to -56 ‰ range, also supported a common
SRB origin of those PLFAs (Londry et al., 2004).
Variation partitioning analyses showed that the concentrations of [3]-ladderane, [5]-
ladderane, 10MeC16:0 and i17:1 correlated with $NH_4^+$ concentration (Figure 6C). Many studies
in other aquatic environments showed that the relative importance of the anammox process is
directly related to the availability of $NH_4^+$ (Dalsgaard and Thamdrup, 2002; Kuypers et al.,
2003). Commonly, the breakdown of OM via ammonification or dissimilatory nitrate reduction to
ammonia (DNRA) is presumed the major sources of $NH_4^+$ for anammox (Kartal et al., 2007).
However, the recent discovery of comammox organisms capable of complete nitrification
underlines the complexity of the nitrogen cycle and the variability of ammonium sources for





anammox (van Kessel et al., 2015). The availability of OM is known as an additional important
factor influencing the anammox process. Higher anammox activity has been observed in OM-
poor environments and interpreted as a decrease in competition for $NO_2^-$ by heterotrophic
denitrifiers (Hu et al., 2011). Consistently, high anammox activity was observed in redox zones
associated to sulphate reduction or sulphur oxidation (Mills et al., 2006; Canfield et al., 2010;
Prokopenko et al., 2013; Wenk et al., 2013). In the groundwater of the wells H5.2 and H5.3, the
occurrence of anammox bacteria and SRB supported low groundwater-surface interactions which
likely threatened the availability of generically favourable electron acceptors and labile OM.
**6.    Conclusion**
In this study, we used constrained ordination to evidence environmental variables that
significantly correlated with PLFA relative abundances in groundwater of distinct carbonate-rock
aquifer assemblages. This technique shows that the active subsurface microbial communities
were mainly affected by variations in dissolved $O_2$, $Fe_t$ and $NH_4^+$ concentrations. Variation
portioning identified PLFA-based microbial functional groups that were directly supported by
results of DNA- and RNA-based amplicon sequencing targeting bacterial 16S rRNA genes.
Higher $O_2$ concentration resulted in increased eukaryotic biomass and higher relative fractions of
nitrite oxidizing bacteria (e.g. *Nitrospira moscoviensis*) but impeded anammox bacteria, sulphate-
reducing bacteria and iron reducing bacteria. In anoxic groundwater, concomitant increase of
total iron ($Fe_t$), $HCO_3^-$ and PLFAs abundant in gram-negative bacteria and fungi suggested the
occurrence of active dissimilatory iron-reduction and a possible role of fungi in meditating iron
solubilisation and transport in those aquifer assemblages. The relative abundance of PLFA
derived from anammox bacteria correlated with $NH_4^+$ concentrations, showing the dependence of



the anammox process on the availability of $NH_4^+$. Our study shows that different relationships
among the microbial community structures, estimated based on both the PLFA patterns and 16S
rRNA gene-targeted next generation sequencing, reflected changes in the physiological strategies
of microorganisms related to a decrease in substrate bioavailability and redox potential of the
groundwater.

ACKNOWLEDGMENT
The work has been funded by the Deutsche Forschungsgemeinschaft (DFG) CRC 1076
"AquaDiva". Field work permits were issued by the responsible state environmental offices of
Thüringen. We thank the Hainich CZE site manager and Heiko Minkmar, Falko Gutmann and
Christine Steinhäuser for scientific coordination and the Hainich National Park. Illumina MiSeq
amplicon sequencing was financially supported by the German Center for Integrative
Biodiversity Research (iDiv) Halle-Jena-Leipzig funded by the Deutsche
Forschungsgemeinschaft (FZT 118).

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



Table 1: Well depths, sampling dates and stratigraphic units of the studied monitoring wells.

| well name | aquifer assemblage | well depth* (m) | Sampling dates | Stratigraphic unit | Notes |
|---|---|---|---|---|---|
| H3.1 | HTL | 42.7-46.7 | June 14 | moTK | well almost dry. Pumped only 100L |
| H3.2 | HTU | 15-22 | June, September 14 | moM | well dry in December 14 |
| H4.1 | HTL | 44.5-47.5 | June, September 14 | moTK | well not accessible in December 14 |
| H4.2 | HTU | 8.5-11.5 | June, September 14 | moM | well not accessible in December 14 |
| H4.3 | HTU | 8.5-12.5 | June, September 14 | moM | well not accessible in December 14 |
| H5.1 | HTL | 84-88 | June, September, December 14 | moTK | |
| H5.2 | HTU | 65-69 | June, September, December 14 | moM | |
| H5.3 | HTU | 47-50 | June, September, December 14 | moM | |

*depth of well screen section below surface; HTL: Hainich transect lower aquifer assemblage; HTU: Hainich transect upper aquifer assemblage; moTK: Upper Muschelkalk, Trochitenkalk formation; moM: Upper Muschelkalk, Meissner formation





Table 2: FunPLFA/BactPLFA, G-/G+ and cy17:0/16:1ω7c ratios averaged in the upper aquifer (HTU) and lower aquifer (HTL) and in the anoxic groundwater at location H4 and H5.

| | BactPLFA | std | FunPLFA | std | G- | std | G+ | std | FunPLFA/BactPLFA | std | G+/G- | std | cy17:0/C16ω7c | std |
|---|---|---|---|---|---|---|---|---|---|---|---|---|---|---|
| HTL | 53.4 | 7.1 | 8.6 | 3.2 | 28.3 | 6.7 | 9.4 | 2.9 | 0.2 | 0.1 | 0.4 | 0.2 | 0.2 | 0.1 |
| HTU | 56.2 | 7.8 | 7.6 | 8.5 | 30.0 | 7.7 | 12.4 | 4.7 | 0.1 | 0.2 | 0.4 | 0.2 | 0.0 | 0.0 |
| H4.2/H4.3 | 55.7 | 6.5 | 17.6 | 7.3 | 26.1 | 4.3 | 17.6 | 1.5 | 0.3 | 0.2 | 0.7 | 0.1 | 0.0 | 0.0 |
| H5.2/H5.3 | 60.1 | 6.7 | 1.9 | 2.2 | 34.9 | 7.6 | 10.4 | 3.5 | 0.0 | 0.0 | 0.3 | 0.1 | 0.0 | 0.0 |



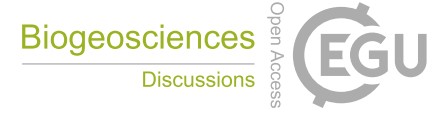

Figure. 1: Schematic representation of the geologic cross section of the Hainich monitoring well transect. The wells sampled for this study are noted in black. The black colours in the wells indicate screen sections and accessed depths of the aquifer assemblages. Abbreviation; mu: Lower Muschelkalk; mm: Middle Muschelkalk; mo: Upper Muschelkalk; moTK: Trochitenkalk formation; moM: Meissner formation; CB: Cycloides-Bank; moW: Warburg formation; ku: Lower Keuper.



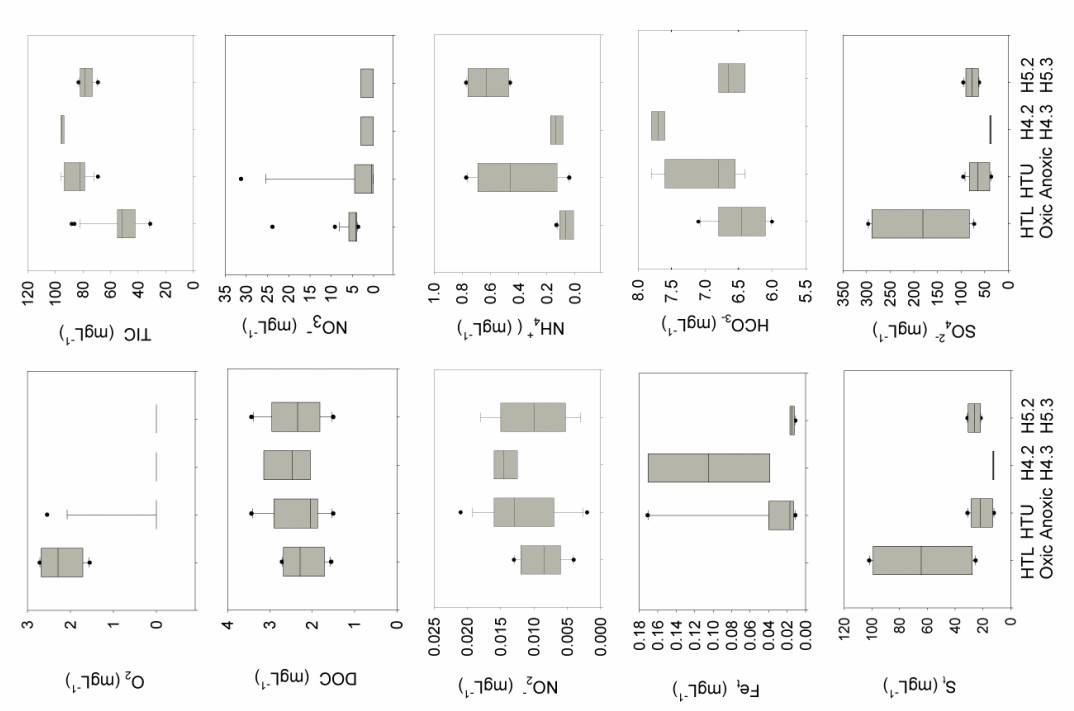

Figure 2: Variations of the chemical compositions of the groundwaters relevant for the discussion. HTL and HTU refer to the wells of the lower and upper aquifer assemblage, respectively. Chemical compositions of the groundwater of the wells H4.2/4.3 and H5.2/5.3 of the HTU are given separately for comparison.





Figure 3: Principal component analysis (PCA) of the groundwater physicochemical compositions. Vectors indicate the steepest increase of the respective physicochemical parameter. The different wells are represented by dots with different colours: blue for oxic groundwater, yellow for sub-oxic groundwater, dark red and violet for anoxic groundwater richer in $Fe_t$ and $NH_4^+$. Note the separation between the lower and upper aquifer (HTL and HTU, respectively) and the anoxic wells at location H4.2/4.3 and H5.2/5.3.



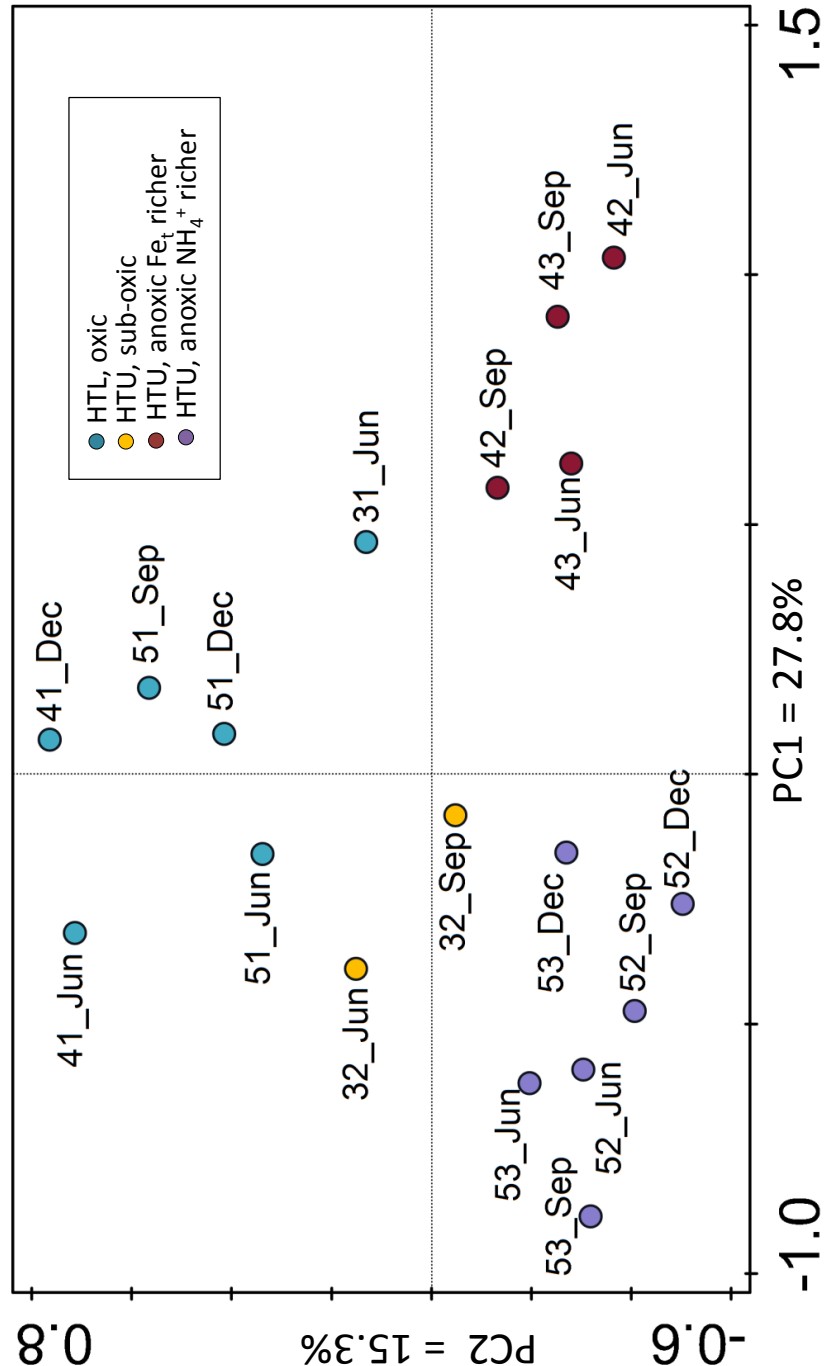

Figure 4: Principal component analysis (PCA) of PLFAs composition. The different wells are represented by dots with different colours: blue for oxic groundwater, yellow for sub-oxic/oxic groundwater, dark red and violet for anoxic groundwater richer in $Fe_t$ and $NH_4^+$. Note the separation between the lower and upper aquifer (HTL and HTU, respectively) and the anoxic wells at location H4.2/4.3 and H5.2/5.3.





Figure 5: Redundancy analysis (RDA) of PLFAs, used as species, and the most significant environmental parameters $O_2$, $NH_4^+$ and $Fe_t$ that explained 37.7% of the variability. The different wells are represented by dots with different colours: blue for oxic groundwater, yellow for sub-oxic groundwater, dark red and violet for anoxic groundwater richer in $Fe_t$ and $NH_4^+$.





Significance test for variation partitioning

| Tested Fraction | % of explained variation | F | P |
|---|---|---|---|
| a+b+c+d+e+f+g | 100 | | |
| a | 19.7 | 2.8 | 0.002 |
| b | 22.0 | 1.9 | 0.012 |
| c | 13.4 | 2.0 | 0.04 |
| a+d | 22.3 | 1.6 | 0.036 |
| b+e | 29.2 | 3.0 | 0.002 |
| c+f | 25.9 | 3.5 | 0.004 |
| | | 2.9 | 0.002 |

Figure 6: Variation partitioning t-value biplots showing the PLFAs significantly correlated with the environmental variables (A) $O_2$, (B) $Fe_t$ and (C) $NH_4^+$. Results of the significance test of the variation partitioning are shown in the associated table. The PLFAs are represented by arrows projecting from the origin. Concentration changes, between sampling data, of a particular PLFA is significantly related to concentration changes of the environmental variables, when the arrow-tip of those PLFA is enclosed within circles. The arrow-tip is enclosed within the red circle for positive correlation and inversely within the blue circle for negative correlation.



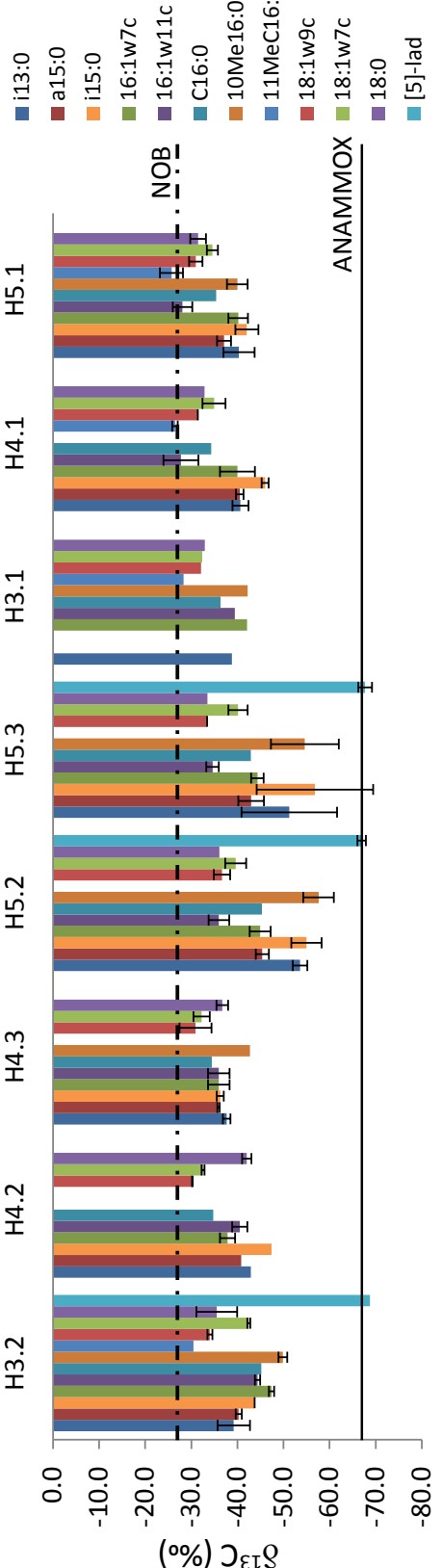

Figure 7: PLFA δ¹³C values averaged in the different wells for June, September and December. The dotted and full lines represent the δ¹³C values of 11MeC16:0 and ladderanes associated with nitrite oxidizing bacteria (e.g. *Nitrospira moscoviensis*) and anammox bacteria, respectively.



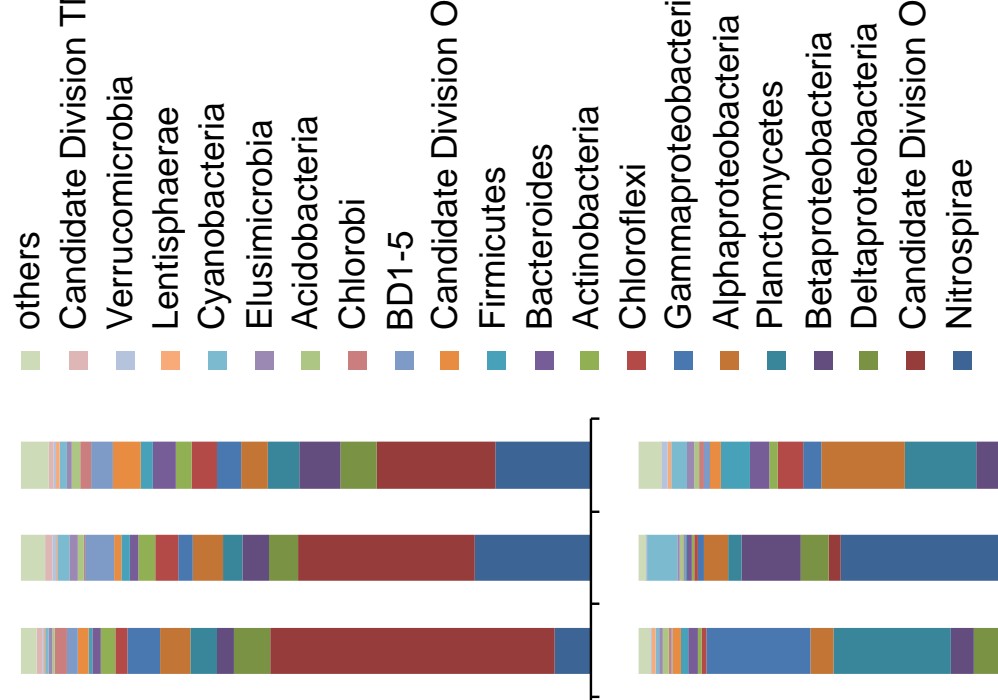

Figure 8: Taxonomic classification (phylum level) of the bacterial communities in the groundwater of the eight different wells from the upper aquifer assemblages (HTU) or the lower aquifer (HTL). (A) Bacterial communities based on sequencing of 16S rRNA genes from extracted genomic DNA. (B) Bacterial communities based on sequencing of RNA-derived 16S rRNA amplicons.



Figure 9: Fractions of sequence reads affiliated with iron oxidizing or iron reducing bacteria, sulfate-reducing bacteria, anammox bacteria, and nitrite oxidizers (*Nitrospira moscoviensis*-related and others) within the bacterial community (A) DNA-based analysis, (B) RNA-based analysis. Only those sequence reads were considered which were unambiguously classified to described taxa on the genus level.