# Peer review of "Functional diversity of microbial communities in pristine aquifers inferred by PLFA- and sequencing -based approaches"

_Biogeosciences, 2016_

## Referee Comment (RC1) · Anonymous Referee #1 · 14 Dec 2016

The MS focuses on analysing the diversity in pristine aquifers via a combination of phospholipid derived fatty acids and DNA and RNA based Illumina amplicon sequencing. The author collected sample in eight different wells which all expressed different mineral, nutrient and redox levels. They showed that the diversity of the wells depend on the biogeochemical composition in those wells. The main issue I have with the research contained in this MS is the following: While the separation of different intact polar lipid classes using a silica column and chloroform (or dichloromethane), acetone and methanol is a commonly used technique since its development in the 50/60s it has been recently shown that the separation is incomplete. Heinzelmann et al. 2014 showed that a considerable portion of the glycolipids end up in the methanol (or so

called phospholipid) fraction and a portion of the phospholipids ends up in the acetone aka glycolipid fraction. Assuming that phospholipids are the main lipids in cell membranes and are considered to reflect living biomass and glycolipids are mainly storage lipids with a longer lifetime after cell death in comparison to phospholipids, this incomplete separation will introduce biases into the data. Therefore, the PLFA fraction described in this MS will not reflect purely membrane lipids and therefore living biomass. Additionally, the fate of SQDGs is completely ignored. SQDGs have been shown to be part of the cell membrane under phosphate limiting conditions. Also, first results indicate that glycolipids play a role as membrane lipids in membranes in phototrophic microorganisms and therefore cannot be considered to be purely storage lipids. This bias has to be addressed in the MS before acceptance. Additionally, especially for someone who isn't too familiar with PCA plots this part of the results/discussions tends to be a bit confusing at times. For they make up a rather big part of the whole paper, please try to make it more clear. Minor comments L18-46: The abstract seems to be quite long, shorten it if possible L68: Not all living organisms have cell membranes consisting of phospholipid derived fatty acids. Cell membranes of archaea consists of etherbound isoprenoid chains and not phospholipid derived fatty acids. Additionally, Van Mooy et al. (2006+2009) showed that under phosphate limited conditions marine microorganisms produce sulphur containing SQDGs instead of phospholipids as membrane lipids. L74-90: This information is quite detailed and could as well be in a table L116: Include the description of the sampling sight into the Material and Methods section L179: In general, the extract obtained during Bligh-Dyer extraction is called Bligh-Dyer-extract or BDE Figure 1: Make it more clear which of the wells are HTU and HTL. I found it a bit confusing. Also make it a bit more clear in the Figure that you didn't sample from H1 and H2 (you never mentioned them in the text but in the figure they suddenly show up). Additionally, make it more clear which well is H3.1 or H3.2 etc.

Technical comments L59: Add an – to SO42 L76: prefixes of fatty acids like iso (i) and anteiso (a or ai) should be in italic L143 and following: Either L or l, please be consequent throughout the text L154-155: Either litre or liter, be consequent throughout

the text L170: I guess you mean CO2 not CO2- L171: Leave a space between mmol and L-1, in general be consequent about having a space between units or not L219: delete the – L280: insert space after - L289: insert space before ± L303: insert space before ± L312: the t in St should subscript L364-367: insert space after ± L402: There is an + missing for NH4+ L403: It is either sulfate or sulphate. Please be consequent throughout the text L413: A dot is missing after the ) L417: Concerning different sulphur species, please stick either to American (f) or British English (ph) L421: Remove the % after 1.1 L439: Subcript the t after Fe L507: Remove the C in 10MeC12:0, check the whole text (be consequent about FA nomenclature throughout the text if you want to add the C or not) L508: Fe2+ not Fe2+ L566: Remove the space before ‰ (check the whole text) L615-933: Reference list is not according to Journal style. Please change that. Also add doi when possible. Number of issues are missing. Also journal names should be abbreviated. L632: space missing after the : L659-660: I assume it is the same author. Please be consequent with the names L665: The 2 has to be in subscript L677 and following: Names of microorganisms have to be in italic L682: space missing before Nitrospira L692-694: Authors initials have to be the same L901: Capital D for Desulfovibrio L907: It is Sinninghe Damsté and not JSS (also check the name in all references for the é) Figure 2: it is HCO3- not HCO3- Figure 3: It is Na

---

## Referee Comment (RC2) · Anonymous Referee #2 · 12 Jan 2017

The authors have carried out an extensive range of chemical and biochemical analyses of pristine aquifer ground water collected in Germany in order to determine microbial diversity. Techniques such as FAME-GC-MS, isotope ratio MS and sequencing showed that the range of microorganisms depended on the groundwater chemistry. This is an interesting study combining both PLFA and PCR data to improve phylogenetic resolution. The follow comments should be addressed before final publication. Line 59 missing 2- on SO4 Line 64: please re-word try replacing "how exactly" with "it is not well understood how" Section 3.3: please describe how external contamination was avoided in the PLFA extractions and analysis. Particularly the 16:0, 16:1, 18:0,18:1 which are almost ubiquitous contaminants. This is particularly important as 16:0 was

shown to be significant in the PLFA distribution analysis. See: Yao, C.-H.; Liu, G.-Y.; Yang, K.; Gross, R.W.; Patti, G.J. Inaccurate quantitation of palmitate in metabolomics and isotope tracer studies due to plastics. Metabolomics 2016, 12 Did the authors check the specificity of the fractions with the SPE method used? Some approaches can see cross contamination with the GL and PL fractions. This could be easily check with standards of GL and PL. Line 184 Is there a reference for the SPE method used? Line 188 define PLOHs ml should be mL throughout Fatty acid quantitation: was there a standard for all fatty acids quantified. I see that a 19:0 fatty acid was used as an internal standard and the Thermo FAME mix as an external standard. Did this contain each FA of that was quantified? If not it is not possible to "quantify" the absolute concentrations of the 47 fatty acids. If there was a standard please state this as it is a key issue for fatty acid quantification. Each FA will have a different response factor. If there wasn't then the mol% cannot be calculated. Peak areas relative to the internal standard could be used for the PCA however.

---

## Author Comment (AC1) · 29 Jan 2017

We thank referee #1 for the constructive and helpful comments. We will carefully consider each of them in the revised manuscript (RM).

Answers general comments:

An important point brings by reviewer 1 and 2 is the separation between the glycolipids and the phospholipids. As mentioned by reviewer 1, an "incomplete separation can result to a significant proportion of glycolipids, betaine lipids and sulfoquinovosyldia- cylglycerols (SQDGs) in the phospholipids fraction. Consequently, the PLFAs fractions might also contain fatty acids derived from glycolipids, betaine lipids, and to some ex-

tent SQDGs, and thus might not only reflect the active biomass", i.e. fatty acids (FA) –derived from phospholipid head groups. As suggested by reviewer 2, we checked the efficiency of the separation by simply running a glycolipid (digalactosyl diglyceride) and a phospholipid (1,2-dinonadecanoyl-sn-glycero-3-phosphatidyl-choline) standard thought the SPE column using the written protocol. After hydrolyze and methylation of the FA, no phospholipid derived FA (C17:0) and glycolipids derived FA (C17:2) was detected in the glycolipids and phospholipids fractions, respectively. We additionally test the glycolipid and phospholipid fractions of the samples for the presence of Glycerol Dialkyl Glycerol Tetratether Lipids (DGDTs). Those lipids could only be detected in the phospholipid fractions. Therefore, it is likely that the phospholipid fractions also contain long chain FA (DGDTs $\geq$ 30 C) derived from glycolipids. However, since in this study we focused on short chain PLFAs ($\leq$ 20 C), we expect these compounds mainly derived from phospholipid head groups and thus represent active organisms. As mentioned by Heinzelmann et al. (2014) an incomplete separation is likely recurrent when using the commonly used PLFA extraction method. Such PLFA extraction/separation protocol has been used for most previous PLFA studies and the development of most PLFA biomarkers. Therefore, in order to be able to compare our study with the previous ones, the similar extraction/separation protocol has been preferred here. The suggestion from Heinzelmann et al. (2014) to study the FA from two fractions (FA eluting in the phospholipid fraction using the common PLFA separation and FA derived from the entire intact polar lipids eluted in a second fraction with MeOH) was not possible, since lipid concentration in such aquifer samples was very low.

However, as asked by referee # 1, this problem will be discussed in the RM. The reader will be informed of such a possible bias. The last part of the introduction will be changed into "Despite PLFAs are widely used in microbial biology, their potential to asses change in microbial structure still remain the topic of much research efforts. A definitive identification of the lipid sources remains often limited because many PLFAs that are commonly associated to a group or genus of bacteria may also be found, albeit in smaller amounts, in cell membranes of other organisms (Frostegård et al.,

2011). Only a few PLFAs are highly specific, for example ladderanes are characteristic membrane constituents of anammox bacteria (Sinninghe Damsté et al., 2005; Sinninghe Damste et al., 2002) and have commonly been used to infer the presence of active anammox bacteria in diverse environments (Kuypers et al., 2003; Jaeschke et al., 2009). As these organisms are capable of anaerobically oxidizing ammonium with nitrite to molecular N2, they play an essential role in N removal from marine (Dalsgaard et al., 2003; Burgin and Hamilton, 2007) and lacustrine environments (Yoshinaga et al., 2011). Yet, their role in aquifer environments is only starting to be considered (Humbert et al., 2009). Another important limitation of PLFA-based studies resides in the fact that the proposed method to separate the glycolipids and phospholipids using a silicic acid column is incomplete and may result in significant proportion of glycolipids, betaine lipids and sulfoquinovosyldiacylglycerols (SQDGs) in the phospholipids fraction (Heinzelmann et al. 2014). Therefore, PLFAs fractions may also contain fatty acids derived from glycolipids, betaine lipids, and to some extent SQDGs, and thus might not only reflect the active biomass. In attempt to overcome these limitations, we combined a detailed multivariate statistical analysis of PLFAs with PLFA $\delta$13C values, and DNA and RNA-based Illumina MiSeq amplicon sequencing of bacterial 16S rRNA genes in groundwaters with very different redox conditions and water chemistry (Kohlhepp et al., 2016). This approach allows parallel study of microbial community composition and specific substrate consumption by evidencing specific PLFAs that respond significantly to change of the groundwater chemistry. Microbial community structure and potential function assessed by PLFAs were confirmed by Illumina MiSeq amplicon sequencing targeting 16S rRNA genes and transcripts, providing a more detailed insight into bacterial community structure and taxonomic affiliation (Kozich et al., 2013). We showed that such PLFA-based study has particular relevance and importance when trying to understand how micro-organisms in groundwater interact with their environment. This study provides baselines for future studies investigating the impact of changes in surface conditions on microorganism in carbonate-rock aquifer ecosystems."

Additionally in the method section, we will add.
"To test the efficiency of the separation between the glycolipids and the phospholipids, the glycolipid standard (digalactosyl diglyceride; Sigma Aldrich) and the phospholipid standards (1,2-dinonadecanoyl-sn-glycero-3-phosphatidyl-choline) were run through the SPE column using the above protocol. The absence of phospholipid derived FA (C17:0) in the glycolipids fraction and glycolipids derived FA (C17:2) in the phospholipids fractions points to an efficient separation and thus a major origin of the studied FAME from phospholipid head groups"

Minor comments referee 1: Text related to PCA analyses will be clarified. L308- 318 "The PCA analyses using the physicochemical parameters of the groundwater separate the wells in three main groups (Figure 3) with 73.6% of the variability explained by the first three principal components (PC): PC1, 32.8%; PC2, 23.8% and PC3, 16.9%. The conductivity, redox potential and the concentrations of Ca2+, SO42-, St and O2 positively correlated (response > 0.5) with PC1 separating the oxic to sub-oxic wells H5.1, H4.1, H3.1 and H3.2 from the anoxic wells H4.2/3 and H5.2/3. The concentrations of NH4+, K+ and Mg2+ inversely correlated (response < 0.5) with PC1, separating wells H5.2/3 from the others. The Fet, TIC and HCO3- positively correlated along PC2 and mainly separated the anoxic wells between location H4 and H5. Groundwaters in location H5 have lower Fet, TIC and HCO3- concentrations but higher NH4+ and K+ concentrations, whereas higher Fet, TIC and HCO3- concentrations but lower NH4+ and K+ concentrations were measured in location H4." And L336-342 "A PCA analysis explained 56.5% of the PLFA variation with PC1 explaining 29.1%; PC2, 15.9% and PC3, 11.5% of overall variability (Figure 4). It separated the wells into the same groups evidenced by PCA analysis of the groundwater chemistry (Figure 3). The wells of the upper aquifer assemblage were separated along PC1; wells from sites H4 separated from those of the sites H5/H3. Along PC2, the wells were separated between the oxic (well H3.1, H4.1 and H5.1), sub-oxic (well H3.2) and anoxic groundwater (H4.2, H4.3, H5.2, H.5.3)."

Abstract will be shortened. L68: thank you for this. This sentence will be changed

to "Phospholipid fatty acids (PLFAs) are important constituents of microbial cell membranes." L74-90: This part will be deleted L116: The description of the sampling site will be included into the Material and Methods. L179: TLE will be changed to BDE

Figure 1: Make it more clear which of the wells are HTU and HTL. I found it a bit confusing. Also make it a bit more clear in the Figure that you didn't sample from H1 and H2 (you never mentioned them in the text but in the figure they suddenly show up). Additionally, make it more clear which well is H3.1 or H3.2 etc.

Figure 1 will be modified to clarify the sampled wells, see attached figure. The following sentence will be added in the RM "Due to very low groundwater level, location H1 and H2 were not sampled."

Technical comments L59: SO42- will be corrected L76: prefixes of fatty acids like iso (i) and anteiso (a or ai) will be in italic L143, L154-L155 L will be noted in the MS L170: Co2- will be corrected L171: We will add a space between the units m mol an and wrote L-1 L219: the – will be deleted L280: space after – L289: space before - L303: insert space before - L312: St will be corrected: t subscript L364-367: space after - L402: NH4+ will be corrected L403: sulphate will be corrected in the MS L413: A dot after the ) will be added L417: sulphur will be corrected in the MS L421: the % after 1.1 will be removed; similar error will be corrected in the MS L439: Fet will be corrected: t: Subcript L507: The C in 10MeC12:0 will be removed and similar error will be corrected in the MS L508: Fe2+ will be corrected L566: The space before % will be remove in the all the MS L615-933: Reference list will be changed according to Journal style. doi will be added when possible. L632: space after the will be added L659-660: Author name will be corrected in the MS L665: The 2 will be noted in subscript L677 Names of microorganisms will be noted in italic L682: space missing before Nitrospira will be added L692-694: Authors initials will be the same L901: Capital D for Desulfovibrio L907: the name Sinninghe Damsté will be corrected Figure 2: HCO3- will be corrected Figure 3: Na will be corrected

[Figure]

[Figure]

[Figure]

modified from Küsel et al. 2016

Figure. 1: Schematic representation of the geologic cross section of the Hainich monitoring well transect. The wells sampled for this study are numbered in black. The black colours in the wells indicate screen sections and accessed depths of the aquifer assemblages. Abbreviation; mu: Lower Muschelkalk; mm: Middle Muschelkalk; mo: Upper Muschelkalk; moTK: Trochitenkalk formation; moM: Meissner formation; CB: Cycloides-Bank; moW: Warburg formation; ku: Lower Keuper.

**Fig. 1.**

---

## Author Comment (AC2) · 29 Jan 2017

We thank referee #2 for the constructive and helpful comments. We will carefully consider each of them in the revised manuscript (RM).

Answer to general comments:

Line 59: S042- will be corrected Line 64: The sentence will be rewritten: "However, how exactly the composition and function of microbial communities in groundwater depend on hydrology, chemistry and the relationship to groundwater recharge dynamics is still not well understood"

Section 3.3: please describe how external contamination was avoided in the PLFA ex-

tractions and analysis. Particularly the 16:0, 16:1, 18:0, 18:1 which are almost ubiqui-tous contaminants. This is particularly important as 16:0 was shown to be significant in the PLFA distribution analysis. See: Yao, C.-H.; Liu, G.-Y.;Yang, K.; Gross, R.W.; Patti, G.J. Inaccurate quantitation of palmitate in metabolomics and isotope tracer studies due to plastics. Metabolomics 2016. Thank you for bringing this point of view: Yes, external contamination was avoided as much as possible. We will add this sentence in the PLFA extraction section "To minimize external contamination, all material (in-cluding filters) and glass in contact with the samples during extraction and purification were baked at 500 °C for 5h to remove organic contaminants. Only trace levels of 16:0 FAME have been detected in blank extracts".

Did the authors check the specificity of the fractions with the SPE method used? Some approaches can see cross contamination with the GL and PL fractions. This could be easily check with standards of GL and PL. Yes, as suggested, we checked the efficiency of the separation by simply running a glycolipid (digalactosyl diglyceride) and a phospholipid (1,2-dinonadecanoyl-sn-glycero-3-phosphatidyl-choline) standard thought the SPE column using the written protocol. No phospholipid derived FA (C17:0) and glycolipids derived FA (C17:2) was detected in the glycolipids and phospholipids fractions, respectively. See also answer to referee 1.

Line 184 Is there a reference for the SPE method used? No.

Line 188 define PLOHs; This will be changed to hydroxy-fatty acids ml should be mL: L will be written in all the RM

Fatty acid quantitation: was there a standard for all fatty acids quantified. I see that a 19:0 fatty acid was used as an internal standard and the Thermo FAME mix as an external standard. Did this contain each FA of that was quantified? If not it is not possible to "quantify" the absolute concentrations of the 47 fatty acids. If there was a standard please state this as it is a key issue for fatty acid quantification. Each FA will have a different response factor. If there wasn't then the mol% cannot be calculated.

[Figure]

Peak areas relative to the internal standard could be used for the PCA however.

Thank you for bringing this important error in view. No commercial standard is available for ladderanes. Therefore, as suggested, we will use the relative peak area for PCA. See attached new PCA figures using relative peak areas (figures 4,5 and 6 of the RM). Changes in the RM will be made accordingly.

―――――――――――――――――――

[Figure]

Figure 4: Principal component analysis (PCA) of PLFAs composition. The different wells are represented by dots with different colours: blue for oxic groundwater, yellow for sub-oxic/oxic groundwater, dark red and violet for anoxic groundwater richer in $Fe_t$ and $NH_4^+$. Note the separation between the lower and upper aquifer (HTL and HTU, respectively) and the anoxic wells at location H4.2/4.3 and H5.2/5.3.

**Fig. 1.**

[Figure]

Figure 5: Redundancy analysis (RDA) of PLFAs, used as species, and the most significant environmental parameters $O_2$, $NH_4^+$ and $Fe_t$ that explained 37.7% of the variability. The different wells are represented by dots with different colours: blue for oxic groundwater, yellow for sub-oxic groundwater, dark red and violet for anoxic groundwater richer in $Fe_t$ and $NH_4^+$.

**Fig. 2.**

[Figure]

Significance test for variation partitioning

| Tested Fraction | % of explained variation | F | P |
|---|---|---|---|
| a+b+c+d+e+f+g | 100 | 3.1 | 0.002 |
| a | 21.0 | 2.2 | 0.008 |
| b | 19.8 | 2.1 | 0.034 |
| c | 16.9 | 1.9 | 0.016 |
| a+d | --- | 3.1 | 0.001 |
| b+e | --- | 3.6 | 0.004 |
| c+f | --- | 3.5 | 0.002 |

Figure 6: Variation partitioning t-value biplots showing the PLFAs significantly correlated with the environmental variables (A) $O_2$, (B) $Fe_t$ and (C) $NH_4^+$. Results of the significance test of the variation partitioning are shown in the associated table. The PLFAs are represented by arrows projecting from the origin. Concentration changes, between sampling data, of a particular PLFA is significantly related to concentration changes of the environmental variables, when the arrow-tip of those PLFA is enclosed within circles. The arrow-tip is enclosed within the red circle for positive correlation and inversely within the blue circle for negative correlation.

**Fig. 3.**

[Figure]

---

## Author Response (AR1)

Dear Dr. Marcel van der Meer,

Please find attached the corrected version of our manuscript entitled "Functional diversity of microbial communities in pristine aquifers inferred by PLFA – and sequencing – based approaches" by Valerie F. Schwab et al.

We found the comments of the referees constructive and helpful, and carefully considered each of them in this revised manuscript (RM). Below please find our point-by-point responses to the referee comments. Changes are reported in yellow background in the RM.

Best regards

Valerie F. Schwab

Answers on the general comments from referee 1:

Possible bias due to an incomplete separation between the glycolipids, betaine lipids, sulfoquinovosyldia-cylglycerols (SQDGs) and phospholipids fractions was mentioned in the introduction lines 83 to 87 and lines 94 to 106.
Additionally, the efficiency of the separation between glycolipids and phospholipids was tested running a glycolipid (digalactosyl diglyceride) and phospholipid (1,2-dinonadecanoyl-sn-glycero-3-phosphatidyl-choline) standard through the SPE column using the written protocol. This was mentioned lines 196-202.

Answers on the minor comments from referee 1:

Text related to PCA analyses was clarified; lines 255-256, lines 312-322 and lines 339-344.
The abstract has been slightly shortened.
Lines 66-67: This sentence has been corrected
Lines 74-90: The description about the possible source of the PLFA was deleted from the introduction and we added Table 1.
The description of the sampling site has been included into the Material and Methods.
Lines 183-184: TLE was changed to BDE
Figure 1 was clarified and the sentence line 122 was added
$SO_4^{2-}$ has been corrected throughout the RM
prefixes of fatty acids like iso and anteiso (a or ai) were put in italic
l was changed to L throughout the RM
$CO_2^-$ was corrected throughout the RM
mol has been changed to % relative abundance (see comments referee 2)
$S_t$ was corrected throughout the RM

$NH_4^+$ was corrected throughout the RM
sulphate and sulphur were corrected throughout the RM
space after % was removed throughout the RM
$Fe_t$ was corrected throughout the RM
The C in 10MeC12:0 (and in other PLFAs) was removed throughout the RM
$Fe^{2+}$ was corrected throughout the RM
Reference list was changed according to Journal style. Doi was added, author name was corrected and name of species was noted in italic. However, I will need indication how I shuo
Figure 2: $HCO_3^-$ was corrected
Figure 3: $Na^+$ was corrected

Answers on the general comments from referee 2:

Line 57: $SO4^{2-}$ has been corrected
Line 62-64 the sentence has been rewritten.
To descript how we avoided external contaminations we added line 176- 178
Results of the test about the efficiency of the separation between the glycolipids and the phospholipids were added line 196-202.
Line 191 PLOHs was changed to hydroxy-fatty acids
ml was changed to mL throughout the MS
As mol% cannot be calculated, as suggested by the referee 2, the PLFAs were quantified relatively the internal standard C19:0 and then converted in % for the PCA analyses. This was noted Line 234-235 and Line 242
% PLFAs have been changed in annexed Tables and PCA and RDA were re-evaluated using % PLFAs (new figures 4,5 and 6).

---

## Author Response (AR2)

Dear Marcel,

Thank you for approving the manuscript and the comments. I also carefully read it again.

Sincerely

Valerie

Line 21: technically I think it are phospholipid derived fatty acids. It might be good to make that clear at least once or twice.
Yes, I introduced this line 21 and 69.

Line 60-61: ….increasing number of studies report the importance of chemolithoautotrophy in groundwater…
Changes done

Line 66: "Intact polar lipids, mainly phospholipids, are important constituents of bacterial and eukaryotic cell membranes and consist of a polar head group linked to a glycerol backbone with two fatty acids esterified to it." Or something similar. The lipids are not in the membrane as fatty acids, but as apolar tails of an intact polar lipid. You have to break the ester bonds to free them, hence the phospholipid derived fatty acids. And it excludes Archaea since they make completely different lipids.

Changes done

Line 77: in general autotrophs are … In the discussion you do mention the reversed TCA cycle and heavy isotopic composition. Would it makes sense here to use something like "typical RuBisCO carbon fixation"?

I preferred to use heterotrophs versus autotrophs since here we theoretically don't have photoautotrophs

Line 83: despite PLFAs being widely used …
Changes done

Line 84: microbial communities
Changes done

Line 85: limitations of PLFA based studies
 Changes done

Line 85-87: The big risk is that so many micro-organisms have never been studied in "pure" culture and we do not know what they make. So I agree with your statement, but it might be even a bit more tricky than that.

And particularly in groundwaters…

Line 94-100: yes, and glycolipids are not only storage lipids, there are also functional glycolipids. So even if the separation would be perfect you would still not necessarily separate structural from storage lipids.

Yes, both DNA and PLFA studies have weakness. I do not think the ideal marker exists. I hope by such a combining approach to overcome some of those problems.

Line 98-99: PLFA fractions
Changes done

Line 178: define FAME, this is now in line 188?
Changes done

Line 184: remove the , after and
Changes done

Line 188: see comment on line 178.
Changes done

Line 283: were instead of was
Change done

Line 338-343: ? I found this confusing, the explanation for PC3 is missing, but there are three "separations"?

The main grouping is along PC1 and PC2 which separated the wells according the water chemistry in three groups. PC3 is not relevant for the discussion since it may separate the wells according the sampling dates. But, this has to be confirmed with more data points. I rewrote this part.

Line 342: either there should be a . after 5.3 or it should be "along". (see also previous comment).
Change done

Line 362 and 363: I assume with increasing O2 concentrations. Perhaps it would be good to actually say that especially in line 363.
I am not sure to understand this comment. I rewrote the sentence.

Line 377: 13C-enriched, more positive and therefore 13C-enriched at least compared to the more negative values associated to Annamox.
Change done

Line 473: eukaryotes such as microalgae etc.
Change done

Line 485: limited in food, for photoautotrophic micro-eukaryotes it might be nutrients, but they also need light.
Photoautotrophic organisms are really rare in groundwater. DNA showed some cyanobacteria but they are likely introduced from surface.

Line 593: you use evidence quite a few times, why not show or sometime suggest. Evidence used in this way feels weird.
I replaced evidence by show.